# Association between sleep duration and hypertension incidence: Systematic review and meta-analysis of cohort studies

Kaveh Hosseini[1], Hamidreza Soleimani[1]*, Kiarash Tavakoli[1,2], Milad Maghsoudi[3], Narges Heydari[3], Yasmin Farahvash[1], Ali Etemadi[1], Kimia Najafi[4], Mani K. Askari[5], Rahul Gupta[6], Diaa Hakim[7], Kazem Rahimi[8,9]

1 Tehran Heart Center, Cardiovascular Diseases Research Institute, Tehran University of Medical Sciences, Tehran, Iran, 2 Students' Scientific Research Center (SSRC), Tehran University of Medical Science, Tehran, Iran, 3 Faculty of Medicine, Isfahan University of Medical Sciences, Isfahan, Iran, 4 Hakim Children Hospital, Tehran University of Medical Science, Tehran, Iran, 5 University of Toledo Medical Center, Toledo, Ohio, United States of America, 6 Lehigh Valley Health Network, Allentown, Pennsylvania, United States of America, 7 Brigham and Women's Hospital, Harvard Medical School, Boston, Massachusetts, United States of America, 8 Deep Medicine, Oxford Martin School, University of Oxford, Oxford, United Kingdom, 9 Nuffield Department of Women's and Reproductive Health, Medical Science Division, University of Oxford, Oxford, United Kingdom

* hamid.r.soleimani90@gmail.com

## Abstract

### Aim

Sleep duration has been suggested to be associated with hypertension (HTN). However, evidence of the nature of the relationship and its direction has been inconsistent. Therefore, we performed a meta-analysis to assess the relationship between sleep duration and risk of HTN incidence, and to distinguish more susceptible populations.

### Methods

PubMed, Embase, Scopus, Web of Science, and ProQuest were searched from January 2000 to May 2023 for cohort studies comparing short and long sleep durations with 7–8 hours of sleep for the risk of HTN incidence. Random-effect model (the DerSimonian-Laird method) was applied to pool risk ratios (RR) and 95% confidence interval (CI).

### Results

We included sixteen studies ranging from 2.4 to 18 years of follow-up duration evaluating HTN incidence in 1,044,035 people. Short sleep duration was significantly associated with a higher risk of developing HTN (HR: 1.07, 95% CI: 1.06–1.09). The association was stronger when the sleep duration was less than 5 hours (HR: 1.11, 95% CI: 1.08–1.14). In contrast to males, females (HR: 1.07, 95% CI: 1.04–1.09) were more vulnerable to developing HTN due to short sleep duration. No significant difference between different follow-up durations and age subgroups was observed. Long sleep duration was not associated with an increased incidence of HTN.

**Data Availability Statement:** All raw data required to replicate the results of our study are within the manuscript and its Supporting information files.

**Funding:** The author(s) received no specific funding for this work.

**Competing interests:** The authors have declared that no competing interests exist.

## Conclusion

Short sleep duration was associated with higher risk of HTN incidence, however, there was no association between long sleep duration and incidence of HTN. These findings highlight the importance of implementing target-specific preventive and interventional strategies for vulnerable populations with short sleep duration to reduce the risk of HTN.

## Introduction

Hypertension (HTN) is the leading preventable risk factor for numerous health conditions, particularly cardiovascular diseases, accounting for 10.8 million deaths and 235 million DALYs (Disability Adjusted Life Years) based on the reports in 2019 [1]. Despite the improvements in knowledge, management, and clinical care of this condition, the prevalence of HTN has grown over the past decade [2]. Therefore, identification of HTN risk factors, early detection of susceptible populations, and timely treatment offers substantial benefits over reduction of HTN burden.

There are many possible behavioral risk factors responsible for HTN incidence including unhealthy diet [3], physical inactivity [4], smoking [5], alcohol consumption [6], and changes in sleeping patterns [7]. Sleep constitutes a significant portion of our daily routine, however, there has been considerable changes in the average sleep duration of population in recent years [8]. About one-third of the adults in the United States reported obtaining less than 7 hours of daily sleep [9].

Both short and long sleep durations are associated to increased risk of major health problems, including diabetes, cardiovascular disease, and mortality [10, 11]. Nevertheless, conflicting information has been published regarding the association between short and long sleep durations and higher risk of HTN incidence. Some studies suggested a U-shaped association between short and long sleep durations and risk of HTN incidence [12, 13], while the others have failed to establish a significant association between long sleep duration and HTN [14–16]. Since the publication of these meta-analyses, other studies have been published in this regard, nevertheless, the exact relations remained unclear.

To address the uncertainties in the existing literature, we conducted a systematic review and meta-analysis of cohort studies to investigate the relationship between varying durations of sleep and risk of HTN incidence. Moreover, to distinguish more vulnerable populations, we also explored the association stratified by age and sex.

## Methods

We adhered to the guidelines outlined in the Preferred Reporting Items for Systematic Reviews and Meta-Analysis (PRISMA) throughout all phases of this meta-analysis [17] (S1 Table). The protocol for this study was registered and published on the International Prospective Register of Systematic Reviews (PROSPERO) (registration number CRD42023434815) on June 23, 2023, prior to data extraction [18]. Due to the lack of data gathering or experimental work within our investigation, our ethical committee deemed to waive the requirement of ethical code for the project, and we suffice to rely on the ethical approval obtained for the included studies.

### Criteria for considering studies for this review

**Type of studies.**  We included cohort studies reporting the incidence of HTN in different sleep durations among the normotensive population. Studies with a sample size of less than 30

people were excluded due to lack of statistical power. Randomized controlled trials (RCTs), case–control studies, cross-sectional studies, review articles, non-English articles, or those lacking substantial information concerning the relationship between sleep duration and HTN incidence were excluded.

**Type of participants and interventions.**   All individuals of both sexes aged 18 years and above, devoid of any prior history of HTN, administration of antihypertensive drugs, or diagnosis of HTN at baseline measurements were included.

## Search methods for identification of studies

A comprehensive search strategy was executed across 5 main databases, including PubMed, Embase, Scopus, Web of Science, and ProQuest, in order to identify pertinent literature from January 2000 to May 2023. Our searches were customized as needed for each database, incorporating relevant keywords and search terms (S2 Table).

## Data collection and management

The results of a systematic search were imported into Endnote software version 20.0 (Clarivate PLC, London, United Kingdom). Two authors (M.M and N.H) independently completed two rounds of screening. The initial screening was based on titles and abstracts, with subsequent full-text review during the second round. Any disparities were resolved through consultation with a third investigator (H.S).

## Risk of bias assessment

Two authors (Y.F and K.T) independently completed a risk of bias assessment for each study utilizing a modified version of the Newcastle-Ottawa Scale tool intended for observational studies [19]. Any discrepancies were checked by the third author (H.S). (S1 Fig).

## Outcomes

Three authors independently extracted data (M.M, N.H and Y.F) while the extracted data checked by a forth author (K.T). The following information was extracted from each eligible study: first author's name, publication year, country of origin, duration of follow-up, demographics; including total sample size, mean age and sex distribution, different methods used to measure sleep duration, reference sleep duration, long and short sleep duration categories, and other covariates used to adjust the reported hazard ratios or risk ratios. Furthermore, hazard ratios (HR) or risk ratios (RR) were harvested to quantify the relationship between sleep duration and the risk of HTN incidence. For each study, we extracted maximally adjusted estimates compared to the referent sleep duration group to reflect the most comprehensive results. When the data of an article was not available, authors were contacted via email for obtaining full data.

## Data analysis

Pooled HRs, RRs, and 95% CI were estimated to quantify the relationship between short sleep duration, long sleep duration, and risk of HTN incidence using random-effect models. Based on the evidence from previous studies, the analyses were performed with sleep durations of 7–8 hours as the reference group [20, 21]. We identified short and long sleep duration by sleeping for less than 7 hours and more than 8 hours, respectively. Heterogeneity was measured using $I^2$ statistics [22]. We conducted a sub-group analysis to evaluate the relationship between sleep duration and risk of HTN incidence after stratification according to the different

definitions of short ($<$ 5 hours, 5 to 7 hours) [23–25] and long (8 to 9 hours, and $>$ 9 hours) [23, 24, 26] sleep durations and different durations of follow-up ($<$ 5 years and $>$5 years), in instances where $I^2$ surpassed 50%. Potential publication bias was evaluated via funnel plot asymmetry, Egger's regression, and the Begg tests [27]. Furthermore, we evaluated the effect of sleep duration and risk of HTN incidence across various age groups and sex categories. All statistical analyses were executed using R Programming language (R for Windows, version 4.1.3, Vienna, Austria) and R Studio version 1.1.463 (Posit PBC, Boston, MA, United States) utilizing the "tidyverse" and "meta" statistical packages.

## Results

### Study characteristics

The initial electronic search retrieved 12,537 articles, among which 4,941 were reviewed based on the title and abstract. A total of 133 full-text articles were retrieved, and 16 studies were included in the final analysis (Fig 1). Inter-readers' agreement was high (Kappa coefficient 0.92). (S3 Table).

The included publications evaluated the relationship between sleep duration and the incidence of HTN in 1,044,035 participants from 6 distinct countries (7 from China, 3 from the USA, 2 from the UK and South Korea, and 1 from Finland and Taiwan). The studies' samples

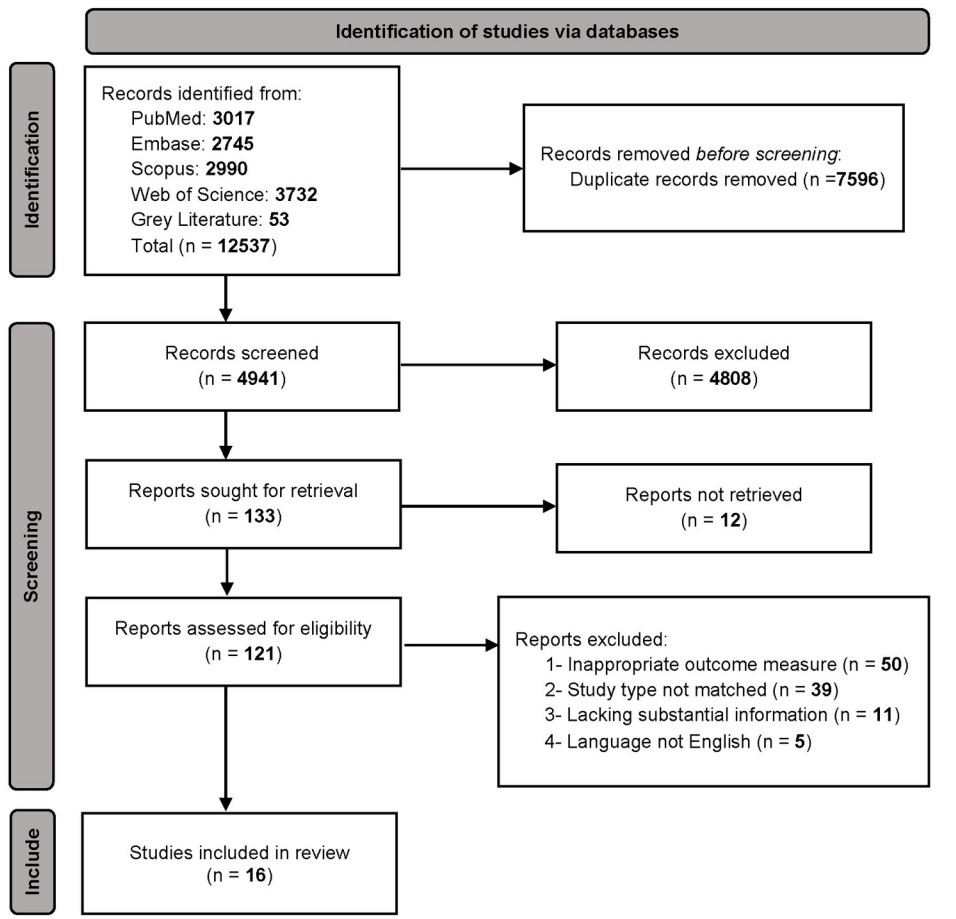

**Fig 1. Study flow chart.**

ranged from 1,009 to 261,279 people. The follow-up duration spanned from 2.4 to 18 years. The mean age of the individuals ranged from 35.4 to 60.9 years. All included studies assessed self-reported sleep duration through questionnaire. The most commonly adjusted confounders included age, smoking, alcohol consumption, sex, physical activity, education, diabetes, depression, marital status, ethnicity, and employment. Most of the studies were based on community-based cohorts; however, 4 studies focused on office employees [21, 28–30], and 1 study centered on hospital nurses [31]. The characteristics of the included studies are detailed in Table 1.

## Short sleep duration and hypertension incidence

Fifteen studies evaluated the impact of short sleep duration (< 7 hours) on the occurrence of HTN [21, 23–26, 29–38]. In the pooled analysis, we observed a statistically significant association between short sleep duration and risk of developing HTN (HR: 1.07, 95% CI: 1.06–1.09). It is important to note that high heterogeneity between studies was detected ($I^2$ = 67%, P < 0.001) (Fig 2A). The preliminary inspection of the funnel plot, Egger's regression, and the Begg tests showed no statistical evidence of publication bias (S2 and S3 Figs).

Upon further stratification of the data, we identified that the association between sleep duration and risk of HTN incidence was more pronounced in people with less than 5 hours of sleep (HR: 1.11, 95% CI: 1.08–1.14, N = 8) compared to people with 5–7 hours of sleep (HR: 1.05, 95% CI: 1–1.10, N = 8) (S4 Fig). Furthermore, the impact of short sleep duration on risk of HTN incidence was predominantly evident in studies with a follow-up duration exceeding 5 years (HR: 1.08, 95% CI: 1.05–1.11, N = 7), whereas no significant association was observed in studies with shorter follow-up durations (HR: 1.03, 95% CI: 0.97–1.10, N = 8) (S5 Fig). However, our analysis found a non-significant test of interaction between follow-up duration and risk of HTN incidence (P value = 0.19). In relation to different subgroups considered, the heterogeneity of effect was not due to the variations in sleep duration or follow-up periods across the included studies. Subgroup analyses of different sleep duration hours and follow-up periods are summarized in Table 2.

The sex-stratified analysis indicated that short sleep duration was associated with higher risk of HTN incidence in females (HR: 1.07, 95% CI: 1.04–1.09, N = 6), whereas this effect was not observed in males (HR: 1.00, 95% CI: 0.96–1.03, N = 4) (P value< 0.01) (Fig 3A). Additionally, our analysis revealed that in contrast to individuals aged 60 and above, short sleep duration might be a risk factor for HTN incidence among individuals under the age of 60 (HR: 1.04, 95% CI: 0.98–1.09, N = 4; HR: 1.07, 95% CI: 1.03–1.11, N = 4; respectively). However, our analysis revealed a non-significant test of interaction between subgroups (P value = 0.4) (Fig 4A).

## Long sleep duration and hypertension incidence

Fourteen studies evaluated the effect of long sleep duration (> 8 hours) on the risk of HTN incidence [21, 23–26, 29–32, 34, 35, 37–39]. In the pooled analysis, we did not discern any significant correlation between long sleep duration and risk of developing HTN (HR: 1.01, 95% CI: 1–1.03). It is noteworthy that our analysis revealed a high degree of heterogeneity among the studies ($I^2$ = 58%, P < 0.001) (Fig 2B). The preliminary inspection of the funnel plot, Egger's regression, and the Begg tests showed no statistical evidence of publication bias (S2 and S3 Figs).

Further stratification of the data unveiled no significant association between HTN incidence and both 8–9 hours of sleep (HR: 1.01, 95% CI: 0.99–1.02, N = 5) and more than 9 hours of sleep (HR: 1.06, 95% CI: 0.99–1.12, N = 11) (S6 Fig). Furthermore, long sleep duration was

**Table 1. Description of the studies included in the meta-analysis.**

| First author (publication year) | Country | Follow-up (years) | Total sample size | Mean age ±SD, range | Gender (Males %) | Referent sleep duration (hours) | Sleep category (hours) | Sleep duration assessment | Adjusted variables |
|---|---|---|---|---|---|---|---|---|---|
| Ganawisch (2006) | USA | 10 | 4,810 | 32–86 years | 36.3 | 7–8 | ≤5<br>6<br>≥9 | Questionnaire | Age, sex, ethnicity, education, diabetes, smoking, overweight/obesity, alcohol consumption, physical activity, salt consumption, depression, daytime sleepiness, pulse rate |
| Kim (2012) | South Korea | 6 | 4,965 | 50.5±8.5 | 46.9 | 5–7 | <5<br>>7 | Questionnaire | Age, sex, body mass index, education, job, income, diabetes, smoking, alcohol consumption, physical activity, snoring, Epworth Sleepiness Scale |
| Ganawisch (2013) | USA | 6 | 60,009<br>32,105<br>68,784 | 30–55<br>30–55<br>25–42 | 0 | 7 | ≤5<br>6<br>8<br>≥9 | Questionnaire | Age, ethnicity, body mass index, diabetes, hypercholesterolemia, smoking, alcohol consumption, caffeine consumption, diet, physical activity, menopause, family history of hypertension, use of aspirin, use of acetaminophen, use of non-aspirin Non-Steroidal Anti-Inflammatory Drugs, snoring, shift work |
| Li (2015) | China | 4.4 | 4,774 | 30–65 | 52.3 | 7-<8 | <6<br>6-<7<br>8-<9<br>≥9 | Questionnaire | Age, sex, waist circumference, education, systolic blood pressure, smoking, alcohol consumption, physical activity, stroke, cardiovascular disease, mental illness, insomnia, psychological pressure, bad mood, use of hypnotics, sleep quality, sleep in daytime, snoring, fasting blood glucose, triglycerides |
| Lu (2015) | China | 4.7 | 1,009 | 35.48±0.19 | 60.7 | <8 | <6<br>6–8 | Questionnaire | Age, sex, education, marital status |
| Clark (2016) | Finland | 4.8 | 70,049 | N/A | 17.27 | 7–8 | <7<br>≥9 | Questionnaire | Age, sex, body mass index, employment, smoking, alcohol consumption, physical activity, respiratory disorders (COPD and asthma), cancer, stress, depression, anxiety |
| Deng (2017) | Taiwan | 18 | 162,121 | 20–80 | 47.4 | 6–8 | <6<br>>8 | Questionnaire | Age, sex, body mass index, waist circumference, education, marital status, systolic blood pressure, smoking, alcohol consumption, physical activity, total cholesterol, triglycerides |
| Song (2016) | China | 3.98 | 32,137 | 46.32 ± 11.50 | 73.4 | 7 | ≤5<br>6<br>8<br>≥9 | Questionnaire | Age, sex, body mass index, systolic blood pressure, diastolic blood pressure, smoking, alcohol consumption, salt consumption, diabetes, hyperlipidemia, physical activity family history of hypertension, resting heart rate |
| Wang (2017) | China | 5 | 9,017 | 60.9 | 50.9 | 7-<8 | <7<br>8-<9<br>9-<10<br>≥10 | Questionnaire | Age, sex, body mass index, education, marital status, shift work, smoking, alcohol consumption, physical activity, tea consumption, caffeine consumption, sleep quality, sleep apnea, use of hypnotics and cardiovascular disease (CVD) drugs, snoring, midday napping, family history of hypertension/chronic diseases, life stress |

*(Continued)*

**Table 1.** (Continued)

| First author (publication year) | Country | Follow-up (years) | Total sample size | Mean age ±SD, range | Gender (Males %) | Referent sleep duration (hours) | Sleep category (hours) | Sleep duration assessment | Adjusted variables |
|---|---|---|---|---|---|---|---|---|---|
| Kim (2018) | South Korea | 2.4 | 106,385 | N/A | 59.1 | 7 | ≤6<br>≥8 | Questionnaire | Age, body mass index, education, marital status, employment, shiftwork, smoking, alcohol consumption, physical activity, diabetes, use of sleeping pills, use of antidepressants, depressive symptom scores, sleep apnea, frequency of snoring, frequency of difficult breathing, sleep quality, family history of hypertension, study center |
| Huang (2021) | China | 5 | 3,178 | 47.98±11.64 | 43 | 8–9 | ≤7<br>≥10 | Questionnaire | Age, sex, body mass index, education, smoking, alcohol consumption, sedentary time per day, family income per capita |
| Li (2021) | UK | 9 | 170,378 | 53.6±8.0 | 36.9 | 7–9 | <7<br>≥9 | Questionnaire | Age, sex, ethnicity, body mass index, education, physical activity, smoking, alcohol consumption, diabetes, depression, cancer, family history of hypertension |
| Lunyera (2021) | USA | 10.1 | 33,497 | 53.9±8.8 | 0 | 7-<9 | <5<br>>5-<7<br>>9 | Questionnaire | Age, ethnicity, body mass index, education, marital status, employment, income level, smoking, alcohol consumption, physical activity, health eating index score, diabetes, use of antidepressants |
| Yao (2021) | China | 4 | 10,176 | 45 and above | 47.6 | 7–9 | ≤5<br>5–7<br>>9 | Questionnaire | Age, sex, education, marital status, employment, smoking, alcohol consumption, physical activity, general obesity, abdominal obesity, diabetes, depression status, neighborhood registration, insurance, housing status and automobile possession, nocturnal sleep duration |
| Yuan (2021) | China | 3 | 9,344 | 36–51 | 42.4 | ≥7 | <5<br>5–6 | Questionnaire | Age, sex, ethnicity, body mass index, education, marital status, employment, smoking, alcohol consumption, diabetes, dyslipidemia |
| Cheng (2022) | UK | 10.9 | 261,297 | 40–69 | 43.4 | 7 | ≤5<br>6<br>8<br>9<br>≥10 | Questionnaire | Age, sex, ethnicity, body mass index, smoking, alcohol consumption, diet, physical activity, family history of cardiometabolic diseases, mental disorders, Townsend deprivation index |

not associated with HTN incidence in studies with a follow-up duration exceeding 5 years (HR: 1.01, 95% CI: 1–1.03, N = 7) or in studies with shorter follow-up durations (HR: 1.02, 95% CI: 0.98–1.05, N = 6) (P value = 0.8) (S7 Fig). In relation to different subgroups considered, the heterogeneity of effect was not due to the variations in sleep duration or follow-up periods across the included studies. Subgroup analyses of different sleep duration hours and follow-up periods are presented in Table 2.

Our analysis revealed that compared to individuals under the age of 60, long sleep duration is correlated with higher risk of HTN incidence in individuals aged 60 and above (HR: 1.04, 95% CI: 0.99–1.09, N = 3; HR: 1.14, 95% CI: 1.03–1.24, N = 3; respectively). However, the test

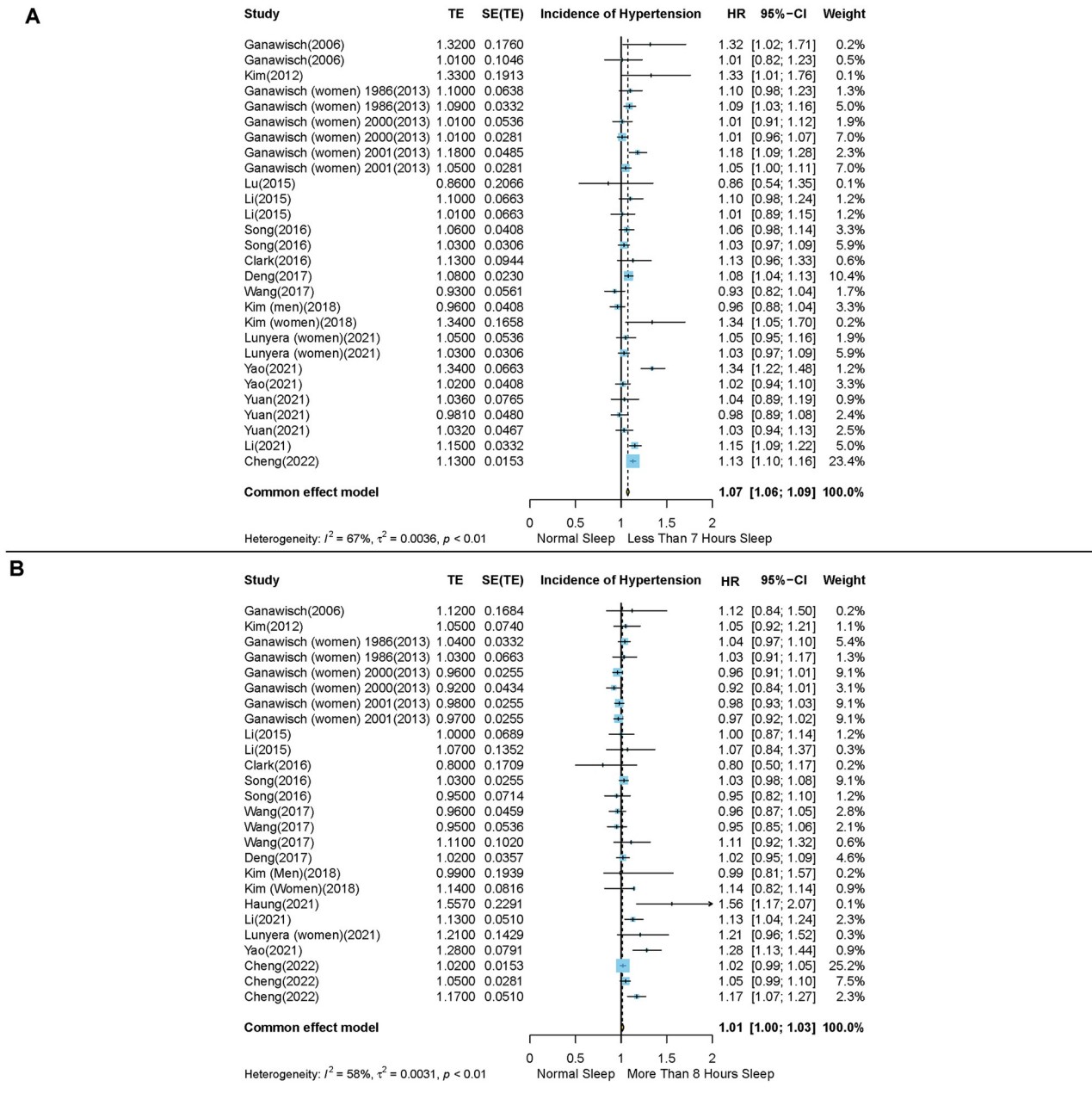

**Fig 2. Forest plots of the association between short and long sleep durations and risk of hypertension incidence.** (A) Short duration of sleep (less than 7 hours) compared with the reference group and (B) Long duration of sleep (more than 8 hours) compared with the reference group. Results are expressed as Hazard ratio and 95% confidence intervals.

of interaction between the two groups was not significant (P value = 0.13) (Fig 4B). Long sleep duration was not associated with higher risk of HTN incidence in either male (HR: 1.02, 95% CI: 0.97–1.06, N = 4) or female (HR: 0.97, 95% CI: 0.99–1.02, N = 6) (P value = 0.36) (Fig 3B).

## Discussion

Based on the results of the present study, short sleep duration, specifically less than 5 hours, is significantly related to increased incidence of HTN. Moreover, females were more susceptible

**Table 2. Subgroup analyses to explore source of heterogeneity.**

| Subgroups | Number of cohorts | HR (95% CI) | I² (%) | P value for heterogeneity |
|---|---|---|---|---|
| *Short sleep duration* | | | | |
| Sleep duration | | | | |
| 5–7 hours | 8 | 1.05(1–1.1) | 80 | <0.01 |
| <5 hours | 8 | 1.11(1.08–1.14) | 63 | <0.01 |
| Duration of follow-up | | | | |
| <5 years | 8 | 1.03(0.97–1.1) | 72 | <0.01 |
| >5 years | 7 | 1.08(1.05–1.11) | 62 | <0.01 |
| *Long sleep duration* | | | | |
| Sleep duration | | | | |
| 8–9 hours | 5 | 1.01(0.99–1.02) | 26 | 0.23 |
| >9 hours | 11 | 1.06(0.99–1.12) | 70 | <0.01 |
| Duration of follow-up | | | | |
| <5 years | 7 | 1.02(0.98–1.05) | 61 | <0.01 |
| >5 years | 6 | 1.01(1–1.03) | 46 | 0.05 |

to this effect. Regarding long sleep duration, no discernible association with HTN incidence was found (Graphical Abstract). This meta-analysis represents a comprehensive investigation into the relationship between sleep duration and risk of HTN incidence, encompassing various definitions of short and long sleep durations, as well as follow-up durations.

## Short sleep duration and hypertension incidence

Short sleep duration was associated with higher burden of HTN incidence. The findings were in line with the results of previous meta-analyses [13, 14, 40], but, it's worth noting that their results were based on a pooled ORs and HRs derived from both cross-sectional and cohort. The OR provides a single snapshot of the association at a certain time point, whereas HR consider both the incidence and the timing and cannot be used interchangeably. Additionally, cohort studies exhibit fewer bias and offer more robust support for causal relationship between different sleep durations and HTN incidence [41–43].

In recent years, three other systematic reviews evaluated the effect of short sleep duration on HTN incidence using cohort studies. Meng et al. [15] and Li et al. [16] conducted meta-analyses covering cohort studies up to 2012 and 2017, respectively. Both studies observed a similar relationship between short sleep duration and risk of HTN incidence, however, they only included 11 and 9 studies and missed some important articles. Che et al. [12] reviewed cohort studies until October 2021 and evaluated the association between sleep duration and the incidence of metabolic syndrome and HTN. Their findings are consistent with the current results but only included 7 cohort studies, featured older data, and omitted many recent studies. Notably, none of these aforementioned reviews evaluated how short sleep duration effects the risk of HTN incidence across various age and sex groups or in different follow-up durations.

Sleep restriction can increase heart rate and blood pressure through various mechanisms including activation of the sympathetic nervous system, or disturbances within the hypothalamic-pituitary-adrenal axis [13, 38]. Of note, our results showed that more follow-up duration might strengthen the effect. This implies that the detrimental effects of inadequate sleep may accumulate over time, leading to a higher risk of HTN.

Based on our findings, it was observed that the female population were more susceptible to HTN incidents when experiencing sleep restriction. These observations align with prior

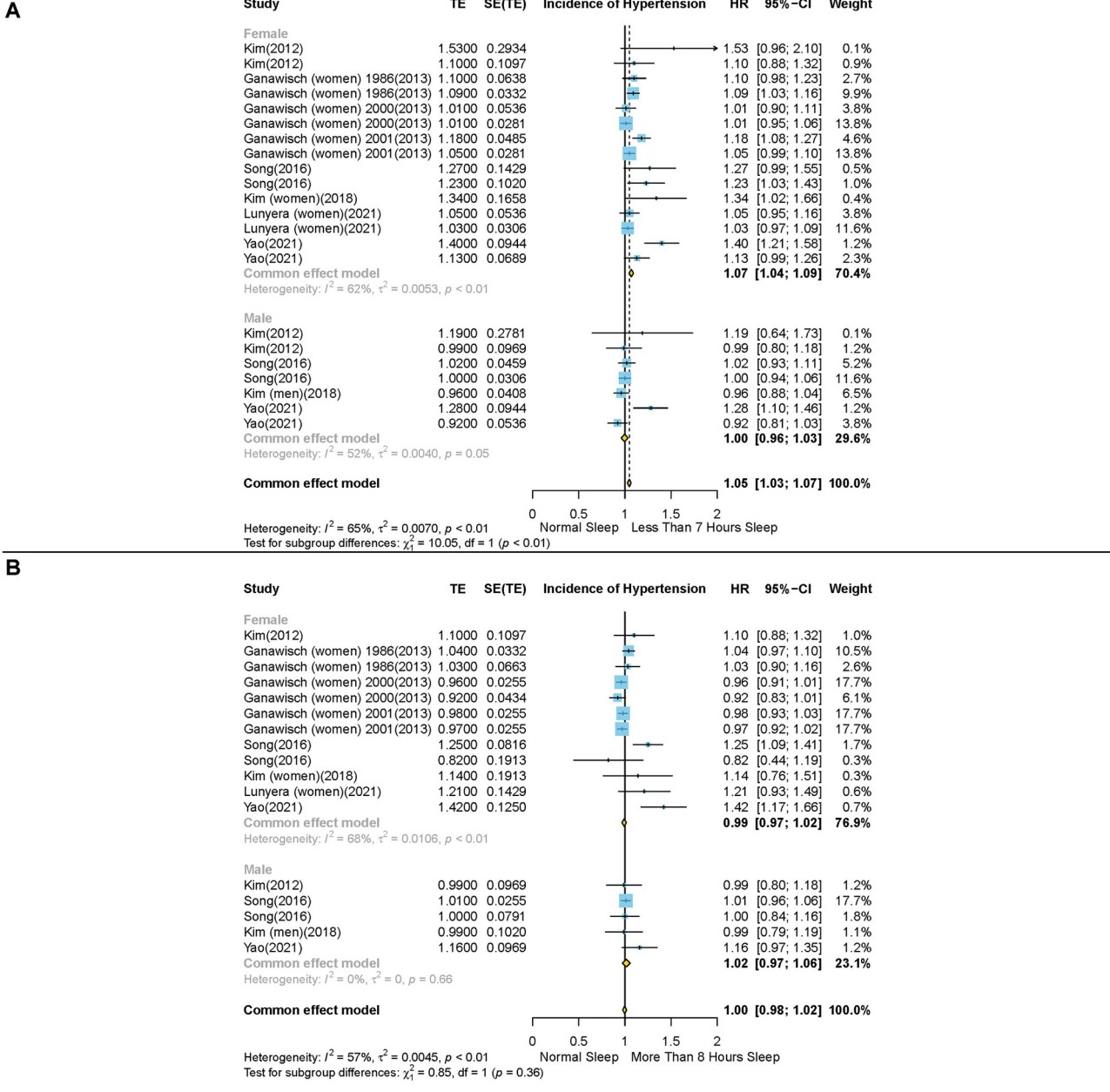

**Fig 3. Forest plots of the association between short and long sleep durations and risk of hypertension incidence in different gender subgroups.**
(A) short duration of sleep (less than 7 hours) compared with the reference group and (B) long duration of sleep (more than 8 hours) compared with the reference group Results are expressed as Hazard ratio and 95% confidence intervals.

studies in the field [40, 44]. The pathophysiology behind the effect of sex on the association between sleep duration and HTN incident is still unclear. Sex differences contribute to variability in the body composition and hormonal profile, influencing the neurohormonal response to sleep patterns [45]. In an experimental study, males showed a more pronounced baroreflex response besides reductions in testosterone levels to diminish the effect of sleep deprivation on HTN, however, females demonstrated a comparatively weaker response in this context [46].

**A**

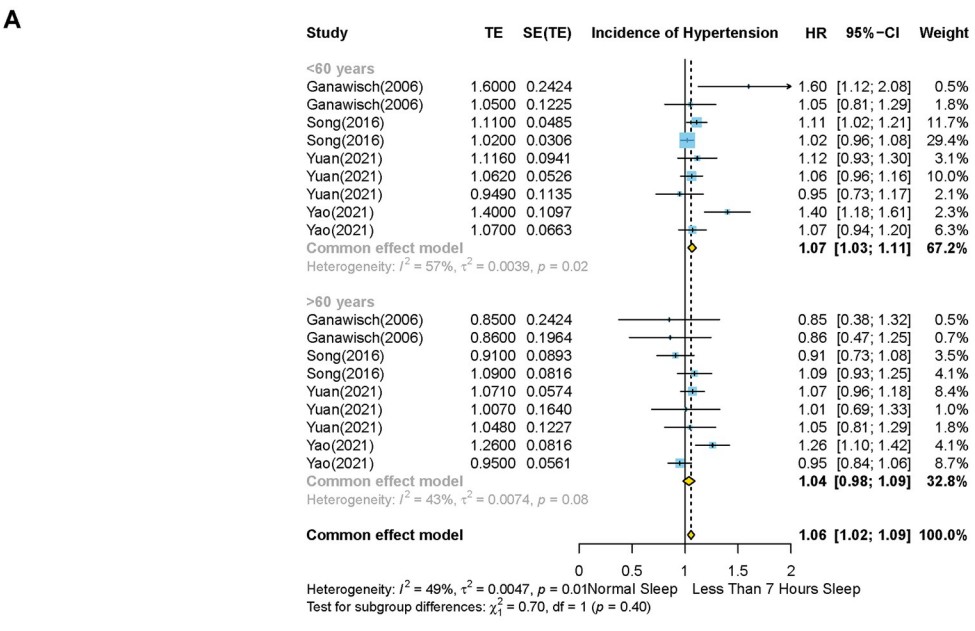

**B**

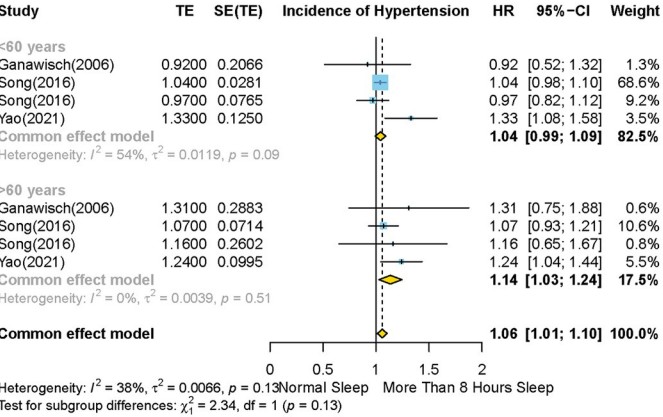

**Fig 4. Forest plots of the association between short and long sleep durations and risk of hypertension incidence in different age subgroups.** (A) short duration of sleep (less than 7 hours) compared with the reference group and (B) long duration of sleep (more than 8 hours) compared with the reference group Results are expressed as Hazard ratio and 95% confidence intervals.

Furthermore, our research suggests that individuals below the age of 60 might be more likely to experience the impact of short sleep duration on the HTN incidence. Elderly individuals, often retired, have more opportunities for daytime napping. These afternoon naps can reverse physiological changes induced by sleep restriction, potentially mitigating HTN risk [24].

## Long sleep duration and hypertension incidence

In contrast to the robust associations between short sleep duration and HTN, our comprehensive analyses revealed a lack of a statistically significant correlation between extended

sleep periods and HTN incidence. The lack of a straightforward association between longer sleep duration and HTN aligns with findings in some previous studies [16, 44]. In contrast, it's worth noting that other studies have shown a significant association between long sleep duration and the incidence of HTN [12, 13]. The relationship between longer sleep duration and HTN incidence in these studies can be attributed to several factors. One plausible explanation is that individuals with sleep disturbances, such as sleep apnea, might unintentionally have longer sleep durations due to their extended sleep periods to compensate for their disrupted sleep patterns, even if their actual sleep duration is short [32]. Moreover, recognized HTN risk factors like depression, and socioeconomic disparities may cause longer sleep durations, which are often overlooked in studies [39]. Nonetheless, the complex association between prolonged sleep and HTN beckons further investigation.

It's worth to note that, our study demonstrated that the incidence of HTN might be more pronounced in individuals older than 60. Age-related changes in vascular structure and function or autonomic dysregulation may contribute to increased blood pressure [47]. Further studies are required to better understand the mechanisms.

As the primary preventable risk factor for cardiovascular diseases, HTN is projected to afflict one third of world population by 2025 [47]. Our study unveiled that less than 7 hours of sleep was associated with a 7% increased risk of developing HTN which escalated to 11% when the sleep duration was less than 5 hours. Therefore, early detection and intervention for individuals with short sleep hold considerable potential for reducing the forthcoming burden of HTN. It would be nice if these findings were supported in large-scale RCTs featuring substantial follow-up duration and more advanced techniques like polysomnography. However, the majority of published RCTs could not add valuable information due to their small sample size and short follow-up durations [16]. An RCT study capable of overcoming these challenges would provide valuable information regarding the relation between sleep duration and HTN incidence. Additionally, there are discrepancies regarding different definitions of short and long sleep duration in published studies. These variations emphasize the need for standardized methodologies in sleep research to enhance the comparability and generalizability of findings across diverse studies.

## Limitations

It is imperative to acknowledge several limitations of our study. First, all data regarding the effect of sleep duration on HTN incidence that are discussed here stem from nonrandomized studies, albeit many of them are prospective in design. Second, the included cohort studies exhibited variability in methodologies, populations, categorization of sleep duration, and adjusted variables which potentially introduce heterogeneity and influence the observed associations. Third, the possibility of publication bias and variations in study quality cannot be entirely ruled out, which may have impacted the overall findings. Fourth, although we employed the results from the most adjusted model in each study, we must acknowledge the potential impact of unmeasured confounding variables on the observed relationships. Finally, the sleep duration in all included studies were assessed by questionnaire, which usually lacks precision to differentiate between time spent asleep from time in bed. Therefore, using other assessment modalities like sleep diaries, actigraphy, and polysomnography can be more useful in evaluating the relationship between sleep duration and risk of HTN incidence. It would be nice if the findings were supported by large-scale randomized trials with more accurate methods in future.

## Conclusion

In conclusion, a significant association between short sleep duration, particularly less than 5 hours, and an elevated risk of HTN was detected. Moreover, females were more vulnerable to this effect. Conversely, long sleep duration showed no significant correlation. These findings underscore the importance of implementing strategies targeting short sleep durations, especially in at-risk populations beside the formulation of additional intervention plans. Further research is warranted to elucidate the underlying mechanisms of the effect of long sleep on HTN and explore potential interventions to mitigate the impact of inadequate sleep on HTN risk.

## Supporting information

**S1 Fig. Risk of bias assessment of the included studies by the Newcastle- Ottawa scale.**
(TIF)

**S2 Fig. Funnel plot for publication bias assessment of studies evaluating hypertension incidence.** (A) short duration of sleep compared with the reference group and (B) long duration of sleep compared with the reference group.
(TIF)

**S3 Fig. Egger and Begg plots for publication bias assessment of studies evaluating hypertension incidence.** (A) short duration of sleep compared with the reference group and (B) long duration of sleep compared with the reference group. (C) short duration of sleep compared with the reference group and (D) long duration of sleep compared with the reference group.
(TIF)

**S4 Fig. Forest plots of the association between hypertension incidence in different definitions of short sleep duration.** (A) 5–7 hours of sleep duration compared with the reference group and (B) less than 5 hours of sleep duration compared with the reference group. Results are expressed as Hazard ratio and 95% confidence intervals.
(TIF)

**S5 Fig. Forest plots of the association between hypertension incidence in different follow-up durations of short sleep.** Results are expressed as Hazard ratio and 95% confidence intervals.
(TIF)

**S6 Fig. Forest plots of the association between hypertension incidence in different definitions of long sleep duration.** (A) 8–9 hours of sleep duration compared with the reference group and (B) more than 9 hours of sleep duration compared with the reference group. Results are expressed as Hazard ratio and 95% confidence intervals.
(TIF)

**S7 Fig. Forest plots of the association between Hypertension incidence in different follow-up durations.** Results are expressed as Hazard ratio and 95% confidence intervals.
(TIF)

**S1 Table. Checklist of compliance with PRISMA guidelines.**
(DOCX)

**S2 Table. Relevant keywords and search terms for searching databases.**
(DOCX)

**S3 Table. Data of the included studies.**
(XLSX)

**S1 Graphical abstract.**
(TIF)

## Acknowledgments

This article does not contain any acknowledgments.

## Author Contributions

**Conceptualization:** Kaveh Hosseini, Hamidreza Soleimani.

**Formal analysis:** Hamidreza Soleimani.

**Project administration:** Kaveh Hosseini, Hamidreza Soleimani.

**Writing – original draft:** Kiarash Tavakoli, Milad Maghsoudi, Narges Heydari, Yasmin Farahvash.

**Writing – review & editing:** Kaveh Hosseini, Hamidreza Soleimani, Kiarash Tavakoli, Ali Etemadi, Kimia Najafi, Mani K. Askari, Rahul Gupta, Diaa Hakim, Kazem Rahimi.

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
