## [Decision Letter · Decision Letter 0]

5 Mar 2024

PONE-D-23-43503Sleep duration and hypertension incidence: Systematic review and meta-analysisPLOS ONE

Dear Dr. Soleimani,

Thank you for submitting your manuscript to PLOS ONE. After careful consideration, we feel that it has merit but does not fully meet PLOS ONE’s publication criteria as it currently stands. Therefore, we invite you to submit a revised version of the manuscript that addresses the points raised during the review process.

We look forward to receiving your revised manuscript.

Kind regards,

Mohammad Hossein Ebrahimi

Academic Editor

PLOS ONE

2. PLOS requires an ORCID iD for the corresponding author in Editorial Manager on papers submitted after December 6th, 2016. Please ensure that you have an ORCID iD and that it is validated in Editorial Manager. To do this, go to ‘Update my Information’ (in the upper left-hand corner of the main menu), and click on the Fetch/Validate link next to the ORCID field. This will take you to the ORCID site and allow you to create a new iD or authenticate a pre-existing iD in Editorial Manager. Please see the following video for instructions on linking an ORCID iD to your Editorial Manager account: https://www.youtube.com/watch?v=_xcclfuvtxQ".

3. We noted in your submission details that a portion of your manuscript may have been presented or published elsewhere. [The article's abstract has been submitted to the "American College of Cardiology 2024" conference; however, we are still awaiting notification regarding its acceptance. It was a conference will not constitute dual publication.] Please clarify whether this conference proceeding was peer-reviewed and formally published. If this work was previously peer-reviewed and published, in the cover letter please provide the reason that this work does not constitute dual publication and should be included in the current manuscript.

5. In the online submission form, you indicated that [The data underlying the results presented in the study are available on request from the author and were gathered from the published studies included in our meta-analysis.]. 

Reviewers' comments:

Reviewer's Responses to Questions

**Comments to the Author**

1. Is the manuscript technically sound, and do the data support the conclusions?

Reviewer #1: Partly

Reviewer #2: Partly

2. Has the statistical analysis been performed appropriately and rigorously? 

Reviewer #1: Yes

Reviewer #2: Yes

3. Have the authors made all data underlying the findings in their manuscript fully available?

Reviewer #1: Yes

Reviewer #2: Yes

4. Is the manuscript presented in an intelligible fashion and written in standard English?

Reviewer #1: Yes

Reviewer #2: Yes

5. Review Comments to the Author

Reviewer #1: 1. Title: The title should be more specific and concise, indicating the focus of the study. For example, "Association Between Sleep Duration and Hypertension Incidence: A Meta-Analysis."

2. Abstract:

• Method: Specify the statistical methods used for pooling hazard ratios and risk ratios (e.g., DerSimonian and Laird random-effects model).

• Results: Line 43 "Ranging from 2.4 to 18 years" might be better as "ranging from 2.4 to 18 years of follow-up duration."

• Conclusion: The conclusion should summarize the main findings of the study and their implications for clinical practice or further research.

3. Introduction:

• When discussing the association between sleep duration and HTN, it would be helpful to briefly mention some of the conflicting findings in the existing literature to highlight the need for the current study.

• The objectives of the study are clearly stated, but it might be beneficial to explicitly mention the primary aim of the meta-analysis (e.g., to quantify the association between sleep duration and HTN incidence) before discussing stratification by age and sex.

• Consider including a brief transition at the end of the introduction to smoothly lead into the methods section, such as mentioning the need for a systematic review and meta-analysis to address the existing uncertainties.

• Line 56-57: Consider rephrasing "With a worldwide prevalence estimated at 34% in men and 32% in women" to specify that these estimates pertain to the prevalence of hypertension.

4. Methods:

• Registration of Protocol: Specify whether the protocol was registered before the initiation of the study. Also, mention the publication date of the protocol if applicable.

• Clarify the rationale behind excluding studies with a sample size of less than 30 people.

• Line 127-129: Please provide references for the definitions of sleep duration utilized in the study.

• When discussing outcomes, providing examples of covariates used for adjustment could enhance understanding. Additionally, specify how missing data were handled during outcome extraction.

5. Discussion:

• Line 226-227: The statement "To our knowledge, this meta-analysis has been the most comprehensive study that exploring the relationship between sleep duration and HTN incidence" could be revised for clarity. For instance, you could say, "This meta-analysis represents a comprehensive investigation into the relationship between sleep duration and HTN incidence, encompassing various definitions of short and long sleep durations, as well as follow-up durations.

• While the discussion references previous meta-analyses and systematic reviews on the topic, it could be strengthened by providing a more detailed comparison of the current findings with those of previous studies. This would help contextualize the significance of the current study and highlight any novel contributions.

• When discussing the association between short sleep duration and HTN incidence, consider elaborating on the potential clinical implications of these findings. For example, you could discuss the importance of early detection and intervention for individuals with short sleep duration to reduce their risk of developing HTN.

• The limitations of the study are appropriately acknowledged in the discussion. To strengthen this section, consider providing a more detailed discussion of how these limitations may have influenced the results and interpretations. Additionally, discuss potential strategies to address these limitations in future research.

• The conclusion effectively summarizes the key findings of the study and emphasizes the implications for clinical practice and future research. To enhance the conclusion, consider reinforcing the main implications of the findings and providing recommendations for future research directions or public health interventions.

Reviewer #2: Hosseini et al. performed a systematic review and meta-analysis to assess the relationship between sleep duration and the risk of incident HTN. I have several minor and major comments for revision:

ABSTRACT

* Please highlight the comparison group in your methods.

* Try to use "risk" before using "incidence" of HTN throughout the manuscript.

* Use pooled "RR" instead of pooled "HR" throughout the manuscript, as you are combining different types of association measures.

INTRODUCTION

* Please state the clinical impressions of your study in the last paragraph of the Introduction. What problems remain unanswered? What are questions responding to?

METHODS

* Please check for consistency with the PRISMA guideline!

* Explain your methods in more detail! How many reviewers contributed to different processes? You should also cite references regarding heterogeneity, Egger's test, etc.

* The NOS assessed quality or risk of bias? These are different!

RESULTS

* Table 1 is not observed completely!

6. PLOS authors have the option to publish the peer review history of their article (what does this mean?). If published, this will include your full peer review and any attached files.

Reviewer #1: **Yes: **Faizul Akmal bin Abdul Rahim

Reviewer #2: No

---

## [Author Response · Author response to Decision Letter 0]

20 Apr 2024

Reviewers' comments

Reviewer #1:

1. Title: 

The title should be more specific and concise, indicating the focus of the study. For example, "Association Between Sleep Duration and Hypertension Incidence: A Meta-Analysis."

Answer: Thank you so much for your valuable comment. We approved the title

2. Abstract: 

• Method: Specify the statistical methods used for pooling hazard ratios and risk ratios (e.g., DerSimonian and Laird random-effects model).

Answer: Thank you so much for your true comment. The text has been changed: line 41-42

• Results: Line 43 "Ranging from 2.4 to 18 years" might be better as "ranging from 2.4 to 18 years of follow-up duration."

Answer: Thank you so much for your insightful comment. The text has been changed: line 43

• Conclusion: The conclusion should summarize the main findings of the study and their implications for clinical practice or further research.

Answer: Your comment has been invaluable. The text has been changed: line 50-53

3. Introduction:

• When discussing the association between sleep duration and HTN, it would be helpful to briefly mention some of the conflicting findings in the existing literature to highlight the need for the current study.

Answer: Thank you so much for your valuable comment. The text has been changed: line 71-78

• The objectives of the study are clearly stated, but it might be beneficial to explicitly mention the primary aim of the meta-analysis (e.g., to quantify the association between sleep duration and HTN incidence) before discussing stratification by age and sex.

Answer: Thank you for sharing your expertise. The text has been changed: line 81

• Consider including a brief transition at the end of the introduction to smoothly lead into the methods section, such as mentioning the need for a systematic review and meta-analysis to address the existing uncertainties.

Answer: Thank you so much for your insightful comment. The text has been changed: line 71-78

• Line 56-57: Consider rephrasing "With a worldwide prevalence estimated at 34% in men and 32% in women" to specify that these estimates pertain to the prevalence of hypertension.

Answer: Thank you so much for your true comment. The text has been changed: line 59-60

4. Methods:

• Registration of Protocol: Specify whether the protocol was registered before the initiation of the study. Also, mention the publication date of the protocol if applicable.

Answer: Thank you so much for your true comment. The text has been changed: line 88

• Clarify the rationale behind excluding studies with a sample size of less than 30 people.

Answer: Thank you so much for your comment. The text has been changed: line 96

• Line 127-129: Please provide references for the definitions of sleep duration utilized in the study.

Answer: Thank you so much for your valuable comment. The text has been changed: line 140-141

• When discussing outcomes, providing examples of covariates used for adjustment could enhance understanding. Additionally, specify how missing data were handled during outcome extraction.

Answer: Thank you so much for your true comment. The text has been changed: line 143 and 162-165

5. Discussion:

• Line 226-227: The statement "To our knowledge, this meta-analysis has been the most comprehensive study that exploring the relationship between sleep duration and HTN incidence" could be revised for clarity. For instance, you could say, "This meta-analysis represents a comprehensive investigation into the relationship between sleep duration and HTN incidence, encompassing various definitions of short and long sleep durations, as well as follow-up durations.

Answer: Thank you so much for the guidance. The text has been changed: line 243-245

• While the discussion references previous meta-analyses and systematic reviews on the topic, it could be strengthened by providing a more detailed comparison of the current findings with those of previous studies. This would help contextualize the significance of the current study and highlight any novel contributions.

Answer: Thank you so much for your constructive comment. The text has been changed: line 248-258

• When discussing the association between short sleep duration and HTN incidence, consider elaborating on the potential clinical implications of these findings. For example, you could discuss the importance of early detection and intervention for individuals with short sleep duration to reduce their risk of developing HTN.

Answer: Thank you so much for your thoughtful comment. The text has been changed: line 299-303

• The limitations of the study are appropriately acknowledged in the discussion. To strengthen this section, consider providing a more detailed discussion of how these limitations may have influenced the results and interpretations. Additionally, discuss potential strategies to address these limitations in future research.

Answer: Thank you for your guidance. The text has been changed: line 319-326

• The conclusion effectively summarizes the key findings of the study and emphasizes the implications for clinical practice and future research. To enhance the conclusion, consider reinforcing the main implications of the findings and providing recommendations for future research directions or public health interventions.

Answer: Thank you so much for your true comment. The text has been changed: line 330-332

Reviewer #2: 

1. Abstract:

* Please highlight the comparison group in your methods.

Answer: Thank you so much for your valuable comment. The text has been changed: line 40

* Try to use "risk" before using "incidence" of HTN throughout the manuscript.

Answer: Thank you so much for your true comment. The text has been changed.

* Use pooled "RR" instead of pooled "HR" throughout the manuscript, as you are combining different types of association measures. 

Answer: All of the included studies in this article used HR as their effect size for assessment, except the study conducted by Haung et al. (https://www.ncbi.nlm.nih.gov/pmc/articles/PMC8517638/ ). Consequently, we kept HR due to its ability to consider both the incidence and the timing beside mitigating the potential confounders, therefore providing stronger support for causal relationship between different sleep durations and HTN incidence. 

2. Introduction:

* Please state the clinical impressions of your study in the last paragraph of the Introduction. What problems remain unanswered? What are questions responding to?

Answer: Thank you so much for your valuable comment. The text has been changed: line 71-78

3. Methods:

* Please check for consistency with the PRISMA guideline!

Answer: Thank you so much for your insightful comment. We rechecked the consistency of the article with PRISMA guidelines and presented the results in S1 table.

* Explain your methods in more detail! How many reviewers contributed to different processes? You should also cite references regarding heterogeneity, Egger's test, etc.

Answer: Thank you so much for your guidance. The method section has been enhanced.

* The NOS assessed quality or risk of bias? These are different!

Answer: Thank you for your true comment. The text has been changed: line 116-117

4. Results:

* Table 1 is not observed completely!

Answer: Thank you for your true comment. The Table 1 has been corrected.

---

## [Decision Letter · Decision Letter 1]

20 Jun 2024

Association between sleep duration and hypertension incidence: Systematic review and meta-analysis of cohort studies

PONE-D-23-43503R1

Dear Dr. Soleimani,

We’re pleased to inform you that your manuscript has been judged scientifically suitable for publication and will be formally accepted for publication once it meets all outstanding technical requirements.

Kind regards,

Mohammad Hossein Ebrahimi

Academic Editor

PLOS ONE

Additional Editor Comments (optional):

Reviewers' comments:

Reviewer's Responses to Questions

**Comments to the Author**

1. If the authors have adequately addressed your comments raised in a previous round of review and you feel that this manuscript is now acceptable for publication, you may indicate that here to bypass the “Comments to the Author” section, enter your conflict of interest statement in the “Confidential to Editor” section, and submit your "Accept" recommendation.

Reviewer #2: All comments have been addressed

Reviewer #3: All comments have been addressed

Reviewer #4: All comments have been addressed

2. Is the manuscript technically sound, and do the data support the conclusions?

Reviewer #2: Yes

Reviewer #3: Yes

Reviewer #4: Yes

3. Has the statistical analysis been performed appropriately and rigorously? 

Reviewer #2: Yes

Reviewer #3: Yes

Reviewer #4: Yes

4. Have the authors made all data underlying the findings in their manuscript fully available?

Reviewer #2: Yes

Reviewer #3: Yes

Reviewer #4: Yes

5. Is the manuscript presented in an intelligible fashion and written in standard English?

Reviewer #2: (No Response)

Reviewer #3: Yes

Reviewer #4: Yes

6. Review Comments to the Author

Reviewer #2: Thanks for your revisions. All comments have been addressed. Your revised manuscript is acceptable now.

Reviewer #3: The manuscript is rather interesting and novel. Well presented and statistical analysis was adequate.

The association between sleep duration and hypertension is clinically significant and short duration of sleep was associated

As expected with increased incidence of htn. The results should be published.

Reviewer #4: Manuscript has been published in JACC already: SLEEP DURATION AND HYPERTENSION INCIDENCE: SYSTEMATIC REVIEW AND META-ANALYSIS

OPEN ACCESS

Prevention And Health Promotion

Aayushi Sood, Kaveh Hosseini, Hamidreza Soleimani, Kiarash Tavakoli, Narges Heydari, Yasmim Fravash, Mani Khorsand Askari, Akshit Sood, Kimia Najafi, Rahul Gupta, Kazem Rahimi, and Diaa A. Hakim

J Am Coll Cardio. 2024 Apr, 83 (13_Supplement) 1877

7. PLOS authors have the option to publish the peer review history of their article (what does this mean?). If published, this will include your full peer review and any attached files.

Reviewer #2: No

Reviewer #3: **Yes: **Dimitrios Afendoulis

Reviewer #4: No

---

## [Editor Report · Acceptance letter]

2 Jul 2024

PONE-D-23-43503R1 

PLOS ONE

Dear Dr. Soleimani, 

I'm pleased to inform you that your manuscript has been deemed suitable for publication in PLOS ONE. Congratulations! Your manuscript is now being handed over to our production team.

Kind regards, 

on behalf of

Dr. Mohammad Hossein Ebrahimi 

Academic Editor

PLOS ONE